# Expression of DAZL Gene in Selected Tissues and Association of Its Polymorphisms with Testicular Size in Hu Sheep

**DOI:** 10.3390/ani10040740

**Published:** 2020-04-23

**Authors:** Zehu Yuan, Jing Luo, Li Wang, Fadi Li, Wanhong Li, Xiangpeng Yue

**Affiliations:** 1State Key Laboratory of Grassland Agro-Ecosystems, Key Laboratory of Grassland Livestock Industry Innovation, Ministry of Agriculture and Rural Affairs, Engineering Research Center of Grassland Industry, Ministry of Education, College of Pastoral Agriculture Science and Technology, Lanzhou University, Lanzhou 730020, China; yuanzh16@lzu.edu.cn (Z.Y.); luoj17@lzu.edu.cn (J.L.); wangl2017@lzu.edu.cn (L.W.); lifd@lzu.edu.cn (F.L.); limh@lzu.edu.cn (W.L.); 2Engineering Laboratory of Sheep Breeding and Reproduction Biotechnology in Gansu Province, Minqin 733300, China

**Keywords:** *DAZL*, expression, Hu sheep, polymorphism, testis

## Abstract

**Simple Summary:**

The deleted in azoospermia-like (*DAZL*) is an RNA binding protein coding gene in autosomal, playing important roles in testicular development and gametogenesis. In this paper, we found that *DAZL* is extremely highly expressed in testis compared with other organs and reaches to a peak at sex maturity (6-month old) in testis. Two single nucleotide polymorphisms (SNPs) within *DAZL* were found to have significant effect on the variation coefficient between left and right epididymis weight.

**Abstract:**

The deleted in azoospermia-like (*DAZL*) gene encoding an RNA binding protein is pivotal in gametogenesis in lots of species and also acts as a pre-meiosis marker. The current study was conducted to detect expression profiles and single nucleotide polymorphisms (SNPs) of *DAZL* in sheep using qPCR, DNA-pooled sequencing, improved multiplex ligase detection reaction (iMLDR^®^) and restriction fragment length polymorphism (RFLP) methods. The results confirmed that ovine *DAZL* showed the highest expression level at six-months of age across five developmental stage. At six-month stage, *DAZL* expressed primarily in testis across seven tissues analyzed. The abundance of *DAZL* in the large-testis group is higher than that in the small-testis group although it is not significant. In addition, six SNPs (SNP1-SNP6) were identified in *DAZL*. Of those, SNP1 (*p* < 0.05) and SNP6 (*p* < 0.01) were significantly correlated with the variation coefficient between left and right epididymis weight (VCTW). The current study implies *DAZL* may play important roles in testicular development and its SNPs are associated with testicular parameters, which supply important indicators for ram selection at early stage.

## 1. Introduction

Since the intensive application of artificial insemination in animal production, males become more important than females. The males with high quality semen not only tend to have more offspring than females, but also help accelerating genetic improvement [1,2]. Therefore, it is worthwhile to select elite rams with high fertility for sheep industry. Traditionally, semen quantity and quality are important indices to evaluate ram fertility. However, semen quantity and quality are very lowly heritable, which can be largely affected by environment, nutrient level, temperature, etc. [3,4,5]. Testis, as a male specific organ functioning in sperm production and androgen secretion, received an increasing amount of attentions in evaluating male fertility. Previous studies have found that testis size is positively correlated with ejaculated volume, sperm motility and sperm density and negatively associated with sperm morphology abnormality [4,5]. Furthermore, testis size is easy to measure and has high heritability (h^2^ = 0.67) [5], which indicates that it is an excellent proxy to select candidate rams with high fertility. Therefore, screening genetic variations in important candidate genes related to testicular development are helpful to select elite rams at early stage.

The deleted in azoospermia (DAZ) family is a kind of RNA binding proteins (RBPs), which plays important roles in spermatogenesis, gametogenesis and postnatal male germ cell development [6,7]. This family comprises of three members that are exclusively expressed in germ cells depending on the specific DAZ members and the species [8], including one Y-chromosomal *DAZ* gene and two autosomal homolog genes: deleted in azoospermia-like (*DAZL*) and boule-like RNA-binding protein (*BOLL,* also known as *BOULE*) [9]. Of those, *DAZL* is conserved throughout the vertebrates with single RNA recognition motif (RRM), while *DAZ* is conserved confined to Old World monkeys, apes and human [8,10]. *DAZL* can be a germ cell marker [11] and showed a declining expression in the testis of human with nonobstructive azoospermia [12]. Additionally, *DAZL* can act as an intrinsic inducer of meiotic initiation in post-migratory germ cells that its expression dramatically increases at the onset of meiosis [13,14]. In *DAZL*-deficient mice, germ cells eventually underwent apoptosis rather than initiating gametogenesis [15]. In human, the single nucleotide polymorphisms (SNPs) within *DAZL* have been found to have significant influence on male fertility of the Chinese population [16].

In sheep, *DAZL* has 11 exons with a length of 22 kb and is located on chromosome 1, encodes a polypeptide chain consisting of 258 amino acids. Li et. al (2020) detected the expression of *DAZL* in testis across several developing stage, finding that *DAZL* showed the highest expression level in the testis after sexual maturity [17]. However, the expression pattern of *DAZL* in Hu sheep at different testicular developmental stages and in different tissues was still unclear. Therefore, the current study detected the expression patterns of *DAZL* in Hu sheep’s heart, liver, spleen, lung, kidney, *longissimus dorsi* and testis and compared its expression in testis among neonatal, pubertal, sexual maturity, body mature and adult stages. In addition, the DNA-pooling sequence, polymerase chain reaction-restriction fragment length polymorphism (PCR-RFLP) and improved multiple-temperature ligase detection reaction (iMLDR^®^) methods were applied to screen SNPs within *DAZL* that associated with testis size and related traits. This study aims to provide theoretical and practical significances for animal breeding and supply molecular markers for marker assisted selection (MAS) at early stage of Hu sheep.

## 2. Materials and Methods

All experimental procedures were carried out following the experimental field management protocols (file No: 2010-1 and 2010-2) approved by Lanzhou University. All efforts were taken to minimize animal suffering.

### 2.1. Animal Management and Sample Collection

Hu sheep, because of high fecundity and perennial estrus, is one of the most widely raised sheep breeds for intensive sheep production system (house feeding) in China. In the current study, a total of 352 healthy male Hu lambs with similar birth weight at similar birth age and without cryptorchidism, were selected from Zhongtian Industry Co. Ltd (Minqin, Gansu, China). This population was divided into 5 groups which were M0 (neonatal, n = 3), M3 (pubertal, n = 3), M6 (sexual maturity, n = 340), M12 (body mature, n = 3) and M24 (adult, n = 3) slaughtered at 0 day, 3 months, 6 months, 12 months and 24 months, respectively. After slaughtered, the testis and epididymis weights and volumes were measured, and samples of testis and epididymis were collected immediately. In addition, 3 Hu sheep closed to the average testis weight at 6 months among all individuals (n = 340) were selected to collect the sample of heart, liver, spleen, lung, kidney and *longissimus dorsi*. All samples were immediately preserved in liquid nitrogen after collection until DNA and RNA extraction.

### 2.2. DNA, Total RNA Extraction and cDNA Synthesis

The genomic DNA was extracted from 340 testis tissues at M6 using a standard phenol-chloroform method [18]. The total RNA was isolated from heart, liver, spleen, lung, kidney, *longissimus dorsi* at M6 (n = 3 for each tissue) and testis at M0, M3, M12, M24 (n = 3, each stage) and M6 (n = 9, three with the largest, smallest and average testis weight, respectively) using RNAsimple Total RNA Kit (TIANGEN Biotech, Beijing, China) following the manufacturer’s protocols. Aliquots (200 ng) of total RNA were used for cDNA synthesis using TransScript One-Step gDNA Removal and cDNA Synthesis SuperMix (TransGen Biotech, Beijing, China) following the manufacturer’s protocols.

### 2.3. Real-Time Quantitative PCR

The expression of *DAZL* in five developmental stages in Hu sheep’s testis, seven different tissues and different size of testis at 6 months was quantified by RT-qPCR. The RT-qPCR was performed on Bio-Rad CFX96 Real-Time PCR System (Bio-Rad Laboratories, Hercules, CA, USA) using 96-well plates. All the operations were followed the MIQE guidelines [19]. Primers used for *DAZL* expression detection were designed using Premier v. 6.0 software (Premier Biosoft Interpairs, Palo Alto, USA) based on ovine *DAZL* sequences (XM_027964130.1) obtained from the NCBI database (Table 1). Each 25-μL real-time qPCR reaction system contained 1 μL of cDNA, 12.5 μL SYBR Premix Ex Taq Perfect Real Time (Takara, Kusatsu, Japan), 0.5 μL of each forward and reverse primer and 10.5 μL of ddH_2_O. The qPCR protocol was as follow: 94 °C for 3 min followed by 40 cycles of 94 °C for 15 s and 60 °C for 30 s, finishing with elongation at 72 °C for 20 s, melting the amplification with constant heating from 65 °C 5 s to 95 °C to obtain the melting curve. All samples were assayed in triplicate. The obtained data were normalized by β-actin and calculated using the 2^-ΔΔCt^ method. Comparisons of gene abundance among five growth stages and seven tissues were implemented by Kruskal–Wallis rank sum test in R 3.6.1 (https://www.r-project.org/). Comparison of gene abundance between large- and small-testis groups was implemented by Wilcoxon Rank Sum test in R 3.6.1.

### 2.4. Sequencing and Genotyping of Ovine DAZL by PCR-RFLP and iMLDR

A total of 11 pairs of primers were designed based on *DAZL* genomic sequence (NC_019458.2) to amplify 10 exons and their flanking regions (Table 1). The genomic DNA was diluted to 50 ng/μL and 10 individuals mixed together as a DNA pool, accounting for 34 DNA pools. Then PCR was conducted using DNA pools as templates. The PCR reaction system was as follow: each 25 μL reaction contained 12.5 μL 2 × Easy Taq SuperMix (TransGen Biotech, Beijing, China), 0.5 μL of each primer (10 pmol/μL), 1 μL genomic DNA-pool (50 ng/μL) and 10.5 μL ddH_2_O. The cycling protocol was 5 min at 95 ℃, 34 cycles of 95 ℃ for 30 s, annealing (Table 1) for 30 s, 72 ℃ for 30–60 s according to the product length (Table 1, 1000 bp/60 s), with a final extension at 72 ℃ for 5 min. The PCR products were sent for sequencing using ABI3730 and sequencing results were analyzed to find putative SNPs using Chromas (Technelysium Pty Ltd., Helensville, Australia). The putative SNPs with 15 bp upstream and downstream sequences were summarized for probes evaluation for SNPs genotyping using improved multiplex ligation detection reaction (iMLDR), which is a newly developed high-throughput SNP genotyping technique (Genesky Biotechnologies, Inc. Shanghai, China). In general, three phosphorylated probes were designed for each SNP locus, including two allele-specific 5′-probes and one shared 3′-probe. Each 5′-probe was consist of a dye-specific sequence at 5′-half and an allele-specific sequence at 3′-half. The 3′-probes was add different size stuffer sequences at the 3′-end to distinguish ligation products with same dye-label. The 3′-end of the allele-specific 5′-probe can hybridize to the target genomic DNA and be ligated with the adjoining 3′-probe and the 5′-end can hybridize to an oligo template and be ligated with a corresponding 5′dye-labeled oligo to make the allele-specific 5′-probe dye-labeling. Hence, the labeling ligation and allelic discrimination ligation produced an effective ligation product. The two allele-specific 5′probes contain different 5′dye-specific sequences that have different 3′-end nucleotide so that they will be labeled with different dyes. To verifying the accuracy of iMLDR, double blind samples were set in the plates, including 6 random replicates of 340 DNA samples genotyped and 2 ddH_2_O samples. In case of probes not available for some SNPs, the RFLP was used to do genotype. The RFLP reaction system was as follow: each 25 μL contained 5 μL PCR products, 0.5 μL restriction enzyme *AluI* (New England Biolabs, Inc., UK), 2.5 μL NEBuffer (1×) and 17 μL ddH_2_O. The system was incubated at 37 ℃ overnight and then visualized in 1% agarose gels.

### 2.5. Statistical Analysis

The testis parameters including variation coefficient between left and right testis weight (VCTW), variation coefficient between left and right epididymis weight (VCEW), testicular index (TI), the ratio of right epididymis weight to right testis weight (REW/RTW), the ratio of left epididymis weight to left testis weight (LEW/LTW), the ratio of total epididymis weight to total testis weight (TEW/TTW) were calculated using Excel, the calculation formula are as follow:(1)$$\mathrm{VCTW}=\frac{\mathrm{standard}\;\mathrm{deviation}\;\mathrm{of}\;\mathrm{left}\;\mathrm{and}\;\mathrm{right}\;\mathrm{testis}\;\mathrm{weight}}{\left(\mathrm{left}\;\mathrm{testis}\;\mathrm{weight}+\mathrm{right}\;\mathrm{testis}\;\mathrm{weight}\right)/2}$$
(2)$$\mathrm{VCEW}=\frac{\mathrm{standard}\;\mathrm{deviation}\;\mathrm{of}\;\mathrm{left}\;\mathrm{and}\;\mathrm{right}\;\mathrm{epididymis}\;\mathrm{weight}}{\left(\mathrm{left}\;\mathrm{epididymis}\;\mathrm{weight}+\mathrm{right}\;\mathrm{epididymis}\;\mathrm{weight}\right)/2}$$
(3)$$\mathrm{TI}=\frac{\mathrm{left}\;\mathrm{testis}\;\mathrm{weight}+\mathrm{right}\;\mathrm{testis}\;\mathrm{weight}}{\mathrm{liveweight}\;\mathrm{before}\;\mathrm{slaughtering}}$$
(4)$$\mathrm{REW}/\mathrm{RTW}=\frac{\mathrm{right}\;\mathrm{epididymis}\;\mathrm{weight}}{\mathrm{right}\;\mathrm{testis}\;\mathrm{weight}}$$
(5)$$\mathrm{LEW}/\mathrm{LTW}=\frac{\mathrm{left}\;\mathrm{epididymis}\;\mathrm{weight}}{\mathrm{left}\;\mathrm{testis}\;\mathrm{weight}}$$
(6)$$\mathrm{TEW}/\mathrm{TTW}=\frac{\mathrm{total}\;\mathrm{epididymis}\;\mathrm{weight}}{\mathrm{total}\;\mathrm{testis}\;\mathrm{weight}}$$

In addition, genotypic, allelic frequencies and genetic parameters—including the polymorphism information content (*PIC*), effective allele number (*Ne*) and expected heterozygosity (*He*), as well as Hardy–Weinberg equilibrium (HWE) testing *p* value—were directly calculated following previous description [20]. Possible outliers (more than four standard deviations from the overall mean) phenotypes were removed for further analysis. The normality of phenotypes were tested by the shapiro.test() function in R 3.6.1. Non-ratio variables (LTV, LTW, LEW, RTV, RTW, REW and TTW) were transform by the sqrt() function and ratio variables (TI, VCTW, VCEW, REW/RTW, LEW/LTW and TEW/TTW) were transformed by the asin(sqrt()) function in R 3.6.1. Association analysis between SNPs and phenotypes was implemented by both ANOVA and Kruskal–Wallis rank sum test in R 3.6.1 (https://www.r-project.org/) fitting a linear model:Y_ijk_ = µ+ G_j_ + E_ijk_(7)
where Y_ijk_ is the trait measured on each of the ijk_th_ animal; µ is the overall population mean; G_j_ is the fixed effect associated with j_th_ genotype; and E_ijk_ is the random error. Differences in mean phenotype among genotypes were tested using the LSD test in agricolae R package (https://cran.r-project.org/web/packages/agricolae/) and Dunn’s Kruskal–Wallis post hoc. *p* < 0.05 was to be considered significant. *p* < 0.01 was considered to be highly significant. Bonferroni correction was applied for multiple testing between genotype groups. The Least Absolute Shrinkage and Selection Operator (LASSO) regression in lasso2 R package (https://cran.r-project.org/web/packages/lasso2/) was used to find the representative SNP. Moreover, the linkage disequilibrium (LD) of identified SNPs was performed using HAPLOVIEW software [21]. The model of association analysis between combined haplotypes and 13 phenotypes was as follows the formula (7) except that the genotype (G) was replaced by combined haplotypes. Numbers less than 12 of the combined haplotypes were excluded from the association analysis.

## 3. Results

### 3.1. The Expression of DAZL in the Testes during Different Growth Stages

The *DAZL* expression in the testes across different growth stages (M0, M3, M6 M12 and M24) was detected by RT-qPCR. Comparison of gene abundance among five growth stages was implemented by Kruskal–Wallis rank sum test. The results revealed that *DAZL* had the highest expression in M6 (Figure 1). The abundance of *DAZL* in M6 was significantly higher than that in M3 (*p* = 0.01). No statistically significant difference was detected between the expression abundance in M6 and M0, in M6 and M12 and in M6 and M24 (*p* > 0.05) (Figure 1). These results implied that the expression abundance of *DAZL* is important for sexual maturity in sheep.

### 3.2. The Expression of DAZL in Different Tissues and Testes with Different Size at M6

The mRNA expression level of *DAZL* was evaluated in different tissue and testes with different size at M6, because of *DAZL* showing the highest expression level at this stage. The samples, including heart, liver, spleen, lung, kidney, *longissimus dorsi* and testis, in addition, three of the biggest (L) and smallest (S) testis from 340 Hu sheep at M6, were selected to compare *DAZL* expression. The result of *DAZL* expression in different tissues showed that *DAZL* expressed extremely high in testis, significantly higher than that in kidney (Figure 2A). The expression of *DAZL* in large-testis group is higher than that in small-testis group (Figure 2B). However, no statistically significant difference was detected between the expression abundance in small-testis group and that in large-testis group (*p* = 0.34).

### 3.3. SNPs Detection and Genotyping

In this study, a total of 6 SNPs were detected by DNA-pooled sequencing (Appendix A), including 1 SNP in exon 10 and 5 SNPs in intronic region (Table 2). Of which, SNP1-SNP5 can be genotyped by iMLDR method, while SNP6 can be recognized by restriction enzyme *AluI* (A^GCT) and was genotyped by PCR-RFLP. Three genotypes of GG, GA and AA were observed in SNP6. Within SNP6 locus, the GG genotype showed a 159 bp band in agarose gel electrophoresis. Heterozygote GA genotype showed three bands of 159 bp, 92 bp and 67 bp. AA genotype showed two bands of 92 bp and 67 bp (Figure 3). The genotypic frequencies of GG, GA and AA were 0.59, 0.37 and 0.04 and G was the dominant allele (0.78). The *He*, *Ne* and *PIC* were 0.35, 1.53 and 0.29 and populations were in line with Hardy–Weinberg equilibrium (Table 3).

The rest 5 SNPs (SNP1~SNP5) were genotyped by iMLDR. All 5 SNPs were successfully genotyped and the genotypic and allele frequency, He, Ne, PIC and Hardy–Weinberg Equilibrium testing *p* value were calculated and summarized in Table 3. The five SNPs were in line with Hardy–Weinberg equilibrium. In addition to SNP2, all the SNPs showed moderate polymorphisms.

### 3.4. Association Analysis between SNPs and Testis Size Parameters

The possible outlier (more than 4 standard deviations from the overall mean) phenotypes were removed for further analysis. The results from Shapiro test indicated that all phenotypes have deviated from normality (Appendix A). In order to make the phenotypes obey the normal distribution as much as possible, non-ratio variable was transformed by sqrt() and the ratio variable was transformed by asin(sqrt()). We using these transform methods because we compared several common transform methods, e.g., sqrt(), asin(), log(x+1), ln(x+1), 1/x; sqrt() and asin(sqrt()) are the best choice for our data. However, some of these transformed phenotypes are still deviated from normality (Appendix A). Thus, both ANOVA and non-parameter methods were implemented for all single SNP association analysis. The results suggested that two statistical methods have consistent results (Table 4). SNP1 had a significant effect on the variation coefficient between left and right epididymis weight (VCEW, *p* < 0.05); SNP6 showed a strong significant association with VCEW (*p* < 0.01). The remaining SNPs had no significant effect on testis parameters (Table 4). Considering the genetic effects for multiple, highly correlated SNPs residing on the same haplotype block are expected biased because the effect(s) are ‘shared’ by many markers, LASSO regression analysis was implemented to identify the representative SNP. However, no significant representative SNP was detected (Appendix A).

### 3.5. Linkage Disequilibrium Analysis

The linkage disequilibrium among the 6 SNPs identified in *DAZL* was calculated by HAPLOVIEW software, as the results showed that all the SNPs were in strong linkage disequilibrium, excepting SNP2 unlinked with SNP1 (D’=1.0, LOD = 0.86, r^2^ = 0.009) and SNP6 (D’ = 1.0, LOD = 1.0, r^2^ = 0.009) (Figure 4). Moreover, SNP1 and SNP4 were in complete linkage disequilibrium (D’ = 1.0, LOD = 37.53, r^2^ = 0.373). In addition, 5 haplotypes were combined. TTCGGG was the domain haplotype with the highest frequency of 0.537, CTTAAA was on the second with the frequency of 0.224, TTTAAG, TGCAAG and CTTAAG were with frequencies of 0.189, 0.027 and 0.014, respectively.

In addition, we also analyzed the genotype haplotypes of 6 SNPs. In total, 4 haplotypes (CTTTCTGAGAGA, CTTTTTAAAAGA, TTTTCCGGGGGG and TTTTCTGAGAGG) with the number greater than 12 were used for further analysis in this population (Table 5). The results showed that the VCEW was significantly different among different haplotypes (Table 5). The VCEW with CTTTCTGAGAGA haplotype is significantly higher than that with TTTTCCGGGGGG and TTTTCTGAGAGG haplotypes.

## 4. Discussion

In this study, the mRNA expression level of *DAZL* was evaluated in testis across five developmental stages, including M0 (neonate), M3 (puberty), M6 (sexual maturity), M12 (body mature) and M24 (adult). The results revealed that *DAZL* showed the highest expression level in testis at M6, which is significantly higher than M3. Our result was in line with previous studies, which found *DAZL* was highly expressed in pubertal (1–1.5 year) and then declined in post-pubertal (2–3 years) and adult (4–8 years) testes of stallion [11]. In contrast, Li et al. (2020) reported that *DAZL* showed the highest expression level in the testes of Tibetan sheep at 12 months of age, which was inconsistent with the current study [22]. This difference may due to that the previous study just compared the expression level of *DAZL* in testes between 5-month-old and 12-month-old, at which Small Tail Han Sheep had not completely reach sexual maturity. The highest expression of *DAZL* in testis at six months in this study maybe consistent with the development of seminiferous tubules and spermatogenesis in the pubertal stage testis according to the study that reported *DAZL* only expressed in the cytoplasm of partial spermatogonia localized within seminiferous tubules at the pre-pubertal stage while in spermatogonia and primary spermatocyte at pubertal stage, in the cytoplasm of spermatogonia at post-pubertal stage [11,23]. This result inclines *DAZL* may mainly function in testicular development and play a key role in regulating spermatogenesis.

At the six-month stage, *DAZL* showed the highest expression level in testis among seven tissues. Meanwhile, large testes had more abundant expression than small testes at this stage though it was not significant. The results also indicated that *DAZL* has important roles in testis development from another point of view. The same results were also found in 12-month-old Small Tailed Han sheep, which revealed that *DAZL* was highly expressed in testis, slightly expressed in heart, while no expression was observed in liver, spleen, lung, kidney and muscle [22]. The previous study evidenced that the testis size had influenced sperm quality which large testis was associate with higher percentage of morphologically normal sperm [5,24]. Hence, we can hypothesis in accordance with the previous study that the *DAZL* has functioned in spermatogenesis [25] that the small testis have defect in spermatogenesis and then incite the reduction of *DAZL* expression. These results may explain the function of *DAZL* in spermatogenesis indirectly, while the possible mechanism that how *DAZL* functioned in spermatogenesis still need further study.

In ram breeding, individuals with large size testis and epididymis are preferred because testis size is positively correlated with ejaculated volume, sperm motility and sperm density and negatively associated with sperm abnormality [4,5]. Therefore, seven phenotypes, including LTV, LTW, LEW, RTV, RTW, REW and TTW which could represent the size of testis or epididymis, were measured in the current study. In addition, elite rams were expected have little difference between left and right sides in terms of testis and epididymis size. For example, the one side or both epididymis enlarge may related with some disease like epididymitis [26] and damage the sperm and may further testicle atrophy [27]. Thus, current study also calculated the VCTW, VCEW, REW/RTW and LEW/LTW. TI and TEW/TTW are expected to be constant in elite dams. The polymorphism of *DAZL* was investigated in this study, six SNPs were found by DNA-pooling sequencing and genotyped by iMLDR and PCR-RFLP. Association analysis was performed between these six SNPs and 13 testicular parameters, finding that SNP1 and SNP6 have a significant association with VCEW. Epididymis where not only the mature of sperm, but also sperm transport, concentration, protection and storage happens, is also an important organ in male reproduction for its function in form vibrant and fertilized sperm [28]. The size of the epididymis is more likely related to sperm competition [29]. The variation coefficient between left and right epididymis weight may be an important trait while it needs further study to prove that.

## 5. Conclusions

In this study, the *DAZL* expression profiles were investigated in Hu sheep, revealing that *DAZL* expressed primarily in testis when compared to other tissues, and large testes had more abundant expression than small testes at six months. Additionally, the expression level of *DAZL* reached to a peak at six-month testis during testicular development. A total of six SNPs were found in *DAZL* which SNP1 and SNP6 had a significant effect on the variation coefficient between left and right epididymis weight. These findings revealed that *DAZL* is crucial during testicular development, and its SNPs which associated with testicular parameters can supply important indicators for ram selection at early stage.

## Figures and Tables

**Figure 1 animals-10-00740-f001:**
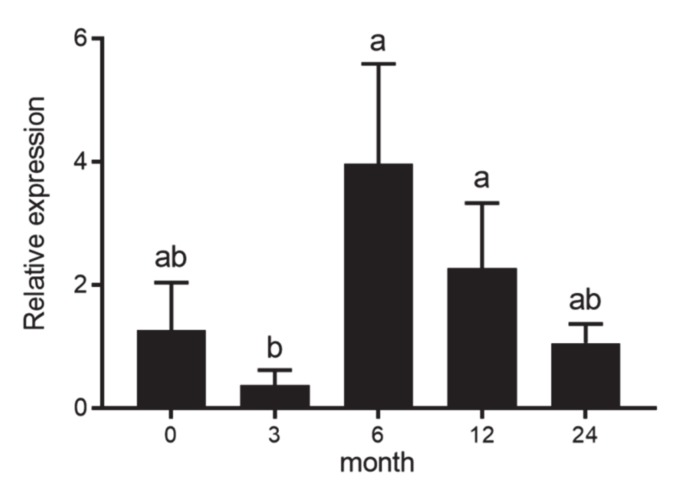
The mRNA expression level of *deleted in azoospermia-like (DAZL)* during testicular development. Different letters above error bars indicate significant differences, *p* < 0.05. Same as below.

**Figure 2 animals-10-00740-f002:**
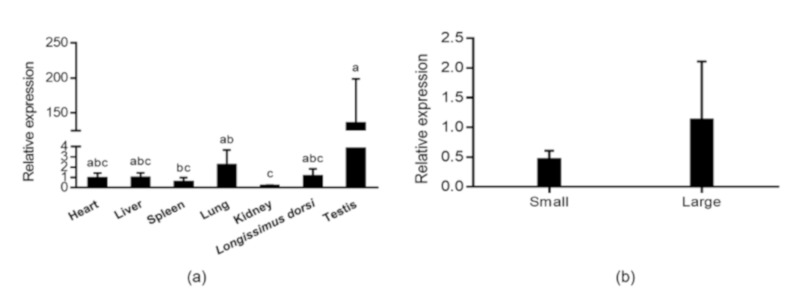
The mRNA expression level of *deleted in azoospermia-like (DAZL)* in different tissue and testes with different size at 6-month old (M6). (**a**): expression level of *DAZL* in different tissues at M6; (**b**): mRNA expression level of *DAZL* in the testes with different size at M6.

**Figure 3 animals-10-00740-f003:**
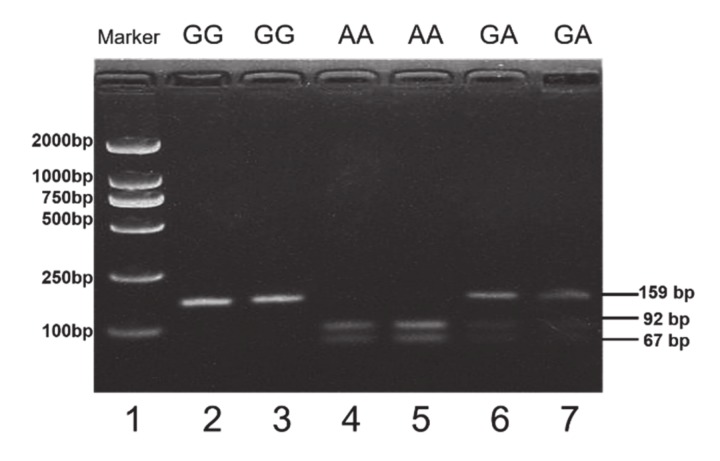
Agarose gel electrophoresis of three genotypes in SNP6 locus within *deleted in azoospermia-like (DAZL ) gene*.

**Figure 4 animals-10-00740-f004:**
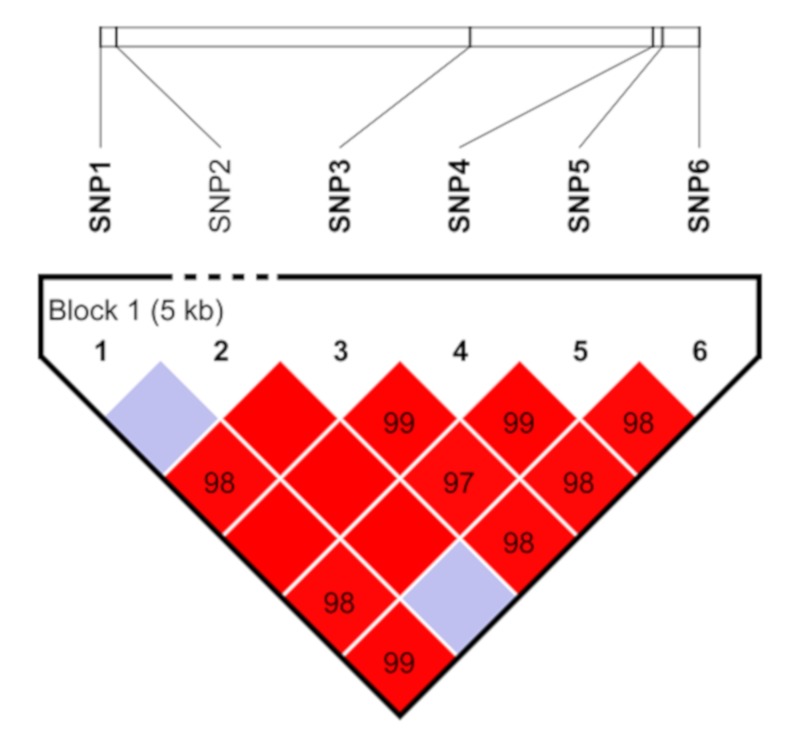
Linkage disequilibrium analysis of 6 single nucleotide polymorphisms (SNPs).

**Table 1 animals-10-00740-t001:** PCR primer sequence information and annealing temperatures for *deleted in azoospermia-like (DAZL)* gene.

Primer	Primer Sequences	Tm (°C)	Product Length (bp)	Targets
DAZL-1	F:5’-ACAGCCTTAACAGAGGTGAATG-3’	64.0	652	Partial of exon 1
	R:5’-GGTAATAATGAGCAGCGGTGAT-3’			
DAZL-2	F:5’-TCACCGCTGCTCATTATTACCT-3’	59.4	835	Partial of exon 1
	R:5’-GTTCCTATTACCTATGCTGATACTGTC-3’			
DAZL-3	F:5’-GACAGTATCAGCATAGGTAATAGGA-3’	64.0	565	Partial of exon 1
	R:5’-AATCACTTGTAGCAGCATCGT-3’			
DAZL-4	F:5’-CTGAGGAGGAGCCACCTAATC-3’	60.5	756	Partial of exon 1
	R:5’-CAAGCACTTCACTTCTCCAACA-3’			
DAZL-5	F:5’-GTCTCTTACTATTCAACACCTGTG-3’	60.5	351	Partial of exon 1 and partial of intron 1
	R:5’-TTTCTGAGTCACCGAGATTTGT-3’			
DAZL-6	F:5’-CTTAACACTCACTCTCAGACTACAG-3’	62.7	261	Exon 2 and partial of intron 2
	R:5’-ATCGGTGGACAGAAGCATACA-3’			
DAZL-7	F:5’-CAGAGGATGGAGTGGCTTCA-3’	62.7	359	Partial of exon 3, intron 3 and exon 4 and partial of intron 4
	R:5’-ATTCTCAGGCACTGGGAAATTC-3’			
DAZL-8	F:5’-GACAGCAAAGGTGAAGACTACAT-3’	60.5	337	Exon 5 and partial of intron 5
	R:5’-GGCTTATCCTCCTTATCCAAGTTC-3’			
DAZL-9	F:5’-CCATCAGTCACAAGTATTCCAACA-3’	60.5	281	Exon 6 and partial of intron 6
	R:5’-TCCTCCTCCTCCACCACAAT-3’			
DAZL-10	F:5’-ACCAGTTCGATCCGTGATTATCT-3’	61.6	268	Exon 7 and partial of intron 7
	R:5’-GTACTTCATGCAGGTTTGGAATTG-3’			
DAZL-11	F:5’-CCTAACATCAATTCCACCAACGA-3’	61.6	302	Exon 9, intron 9 and partial of exon10
	R:5’-GTGATTCATCCATCCCAGCATT-3’			
DAZL	F: 5’-GGCTCCTCCTCAGACATT-3’	60.0	226	mRNA(for qPCR)
	R: 5’-TGCTGCTACAAGTGATTCC-3’			
β-actin	F: 5’-CTGAGATCAGCCGCGATAA-3’	60.0	220	mRNA(for qPCR)
	R: 5’-TTAATGAGCACAAAGTACGT-3’			

F: Forward primer; R: Reverse primer.

**Table 2 animals-10-00740-t002:** Position and mutation types of detected single nucleotide polymorphisms (SNPs)

SNPs	Position	Location	Alleles	Mutation Type
SNP1	c1.g.271493870	intron 4	C/T	intronic
SNP2	c1.g.271494008	intron 4	G/T	intronic
SNP3	c1.g.271497106	intron 5	C/T	intronic
SNP4	c1.g.271498705	intron 9	G/A	intronic
SNP5	c1.g.271498790	intron 9	G/A	intronic
SNP6	c1.g.271499112	exon 10	G/A	synonymous

**Table 3 animals-10-00740-t003:** Genotypic and allelic frequencies, expected heterozygosity (He), effective allele number (Ne), polymorphism information content (PIC), Hardy–Weinberg equilibrium (HWE) testing *p* value of single nucleotide polymorphisms (SNPs) identified in *DAZL*.

SNPs	Genotypic Frequency			Allelic Frequency		*He*	*Ne*	*PIC*	HWE *p* value
SNP1	TT	CT	CC	C	T	0.37	1.58	0.30	0.82
	0.57	0.38	0.05	0.24	0.76				
SNP2	TT	GT	GG	G	T	0.06	1.06	0.05	1.00
	0.94	0.06	0.00	0.03	0.97				
SNP3	CC	CT	TT	C	T	0.49	1.97	0.37	1.00
	0.32	0.49	0.19	0.57	0.43				
SNP4	GG	GA	AA	G	A	0.50	1.99	0.37	0.89
	0.29	0.51	0.20	0.54	0.46				
SNP5	GG	GA	AA	G	A	0.50	1.99	0.37	0.87
	0.29	0.51	0.20	0.54	0.46				
SNP6	GG	GA	AA	G	A	0.35	1.53	0.29	0.29
	0.59	0.37	0.04	0.78	0.22				

**Table 4 animals-10-00740-t004:** Association analysis between single nucleotide polymorphisms within *deleted in azoospermia-like* (*DAZL*) gene and testicular traits.

SNPs	Genotype	N	LTV ^1^	LTW ^2^	LEW ^3^	RTV ^4^	RTW ^5^	REW ^6^	TTW ^7^	TI ^8^ (‰)	VCTW ^9^ (%)	VCEW ^10^ (%)	REW/ RTW ^11^	LEW/ LTW ^12^	TEW/ TTW ^13^
SNP1	CC	16	117.5 ± 49.93	116.27 ± 44.78	17.34 ± 3.63	115.33 ± 53.57	112.74 ± 47.1	15.97 ± 3.14	244.43 ± 107.56	6.09 ± 2.78	3.40 ± 3.63	6.17^ab^ ± 5.37	0.16 ± 0.04	0.16 ± 0.05	0.16 ± 0.04
	CT	112	119.38 ± 50.79	116.24 ± 48.32	17.87 ± 5.33	116.1 ± 48.3	113.89 ± 47.83	16.66 ± 4.68	230.13 ± 95.46	5.85 ± 2.51	3.91 ± 4.00	6.48^a^ ± 5.86	0.16 ± 0.06	0.17 ± 0.06	0.16 ± 0.05
	TT	171	109.91 ± 44.8	109.24 ± 43.47	17.62 ± 4.8	110.01 ± 43.81	107.66 ± 43.78	16.91 ± 4.45	216.90 ± 86.68	5.72 ± 2.32	3.10 ± 2.91	4.60^b^ ± 3.86	0.18 ± 0.07	0.18 ± 0.07	0.18 ± 0.07
ANOVA P			0.299	0.466	0.898	0.615	0.553	0.724	0.366	0.860	0.330	0.010	0.277	0.290	0.149
Kruskal test P			0.395	0.564	0.984	0.669	0.639	0.779	0.507	0.928	0.695	0.023	0.268	0.367	0.269
SNP2	GT	17	111.47 ± 56.81	109.77 ± 57.81	18.66 ± 6.19	111.53 ± 57.4	107.34 ± 55.43	17.68 ± 5.72	217.11 ± 112.72	5.20 ± 2.42	3.00 ± 1.85	0.05 ± 0.04	0.19 ± 0.07	0.19 ± 0.07	0.19 ± 0.07
	TT	282	114.02 ± 46.96	112.39 ± 44.66	17.64 ± 4.86	112.62 ± 45.32	110.44 ± 44.89	16.71 ± 4.4	223.70 ± 90.03	5.82 ± 2.41	3.44 ± 3.48	0.05 ± 0.05	0.17 ± 0.07	0.18 ± 0.07	0.17 ± 0.06
ANOVA P			0.749	0.690	0.458	0.799	0.680	0.424	0.657	0.322	0.983	0.865	0.169	0.236	0.141
Wilcoxon test			0.621	0.539	0.637	0.696	0.587	0.757	0.564	0.251	0.601	0.973	0.155	0.226	0.175
SNP3	CC	96	110.11 ± 44.76	109.47 ± 43.64	18.02 ± 5.08	110.32 ± 44.16	108.69 ± 43.33	17.52 ± 4.53	218.16 ± 86.33	5.78 ± 2.21	3.39 ± 2.87	4.44 ± 3.71	0.18 ± 0.07	0.18 ± 0.07	0.18 ± 0.06
	CT	147	114.6 ± 48.24	112.3 ± 45.71	17.79 ± 5.03	112.9 ± 45.31	109.95 ± 45.79	16.59 ± 4.32	222.25 ± 90.87	5.78 ± 2.47	3.48 ± 3.70	5.87 ± 5.62	0.17 ± 0.07	0.18 ± 0.07	0.17 ± 0.06
	TT	56	118.39 ± 50.18	116.81 ± 47.88	16.91 ± 4.44	115.53 ± 51.26	113.83 ± 48.72	15.93 ± 4.66	235.02 ± 100.69	5.83 ± 2.64	3.30 ± 3.52	5.76 ± 4.28	0.16 ± 0.05	0.16 ± 0.05	0.17 ± 0.05
ANOVA P			0.619	0.668	0.455	0.862	0.833	0.077	0.609	0.980	0.810	0.093	0.181	0.145	0.102
Kruskal test P			0.681	0.711	0.253	0.855	0.894	0.082	0.721	0.986	0.385	0.117	0.114	0.130	0.120
SNP4	AA	62	117.02 ± 48.89	115.14 ± 46.97	17.07 ± 4.43	113.84 ± 50.16	112.06 ± 47.79	16.07 ± 4.5	231.19 ± 98.68	5.73 ± 2.53	3.19 ± 3.38	5.84 ± 4.24	0.16 ± 0.05	0.17 ± 0.05	0.16 ± 0.05
	GA	151	115.17 ± 49.82	113.08 ± 47.36	17.9 ± 5.26	114.07 ± 46.87	110.95 ± 47.13	16.75 ± 4.66	224.03 ± 93.82	5.78 ± 2.49	3.56 ± 3.66	5.65 ± 5.58	0.17 ± 0.07	0.18 ± 0.07	0.17 ± 0.06
	GG	86	109.29 ± 42.03	108.66 ± 40.78	17.81 ± 4.73	109.01 ± 41.43	107.77 ± 40.95	17.29 ± 4.11	216.43 ± 81.11	5.84 ± 2.19	3.33 ± 2.97	4.62 ± 3.78	0.18 ± 0.07	0.18 ± 0.07	0.18 ± 0.06
ANOVA P			0.651	0.738	0.594	0.803	0.904	0.224	0.718	0.895	0.797	0.250	0.422	0.319	0.287
Kruskal test P			0.730	0.794	0.423	0.761	0.899	0.162	0.797	0.890	0.665	0.227	0.279	0.310	0.283
SNP5	AA	61	117.46 ± 49.17	115.56 ± 47.24	17 ± 4.43	114.23 ± 50.48	112.55 ± 48.04	16.01 ± 4.51	232.16 ± 99.2	5.77 ± 2.55	3.19 ± 3.41	5.86 ± 4.27	0.16 ± 0.05	0.16 ± 0.05	0.16 ± 0.05
	GA	151	115.27 ± 49.93	113.21 ± 47.54	17.91 ± 5.27	114.3 ± 47.03	111.17 ± 47.37	16.78 ± 4.66	224.39 ± 94.25	5.79 ± 2.53	3.53 ± 3.64	5.61 ± 5.52	0.17 ± 0.07	0.18 ± 0.07	0.17 ± 0.06
	GG	87	108.9 ± 41.59	108.21 ± 40.21	17.83 ± 4.71	108.39 ± 40.87	107.09 ± 40.28	17.27 ± 4.08	215.3 ± 79.83	5.80 ± 2.12	3.39 ± 3.00	4.69 ± 3.93	0.18 ± 0.08	0.18 ± 0.07	0.18 ± 0.06
ANOVA P			0.595	0.680	0.520	0.730	0.841	0.209	0.650	0.960	0.755	0.286	0.342	0.246	0.216
Kruskal test P			0.669	0.730	0.337	0.677	0.820	0.144	0.723	0.950	0.551	0.242	0.205	0.213	0.198
SNP6	AA	11	108.18 ± 38.49	108.02 ± 34.09	17.16 ± 3.46	108 ± 43.79	104.99 ± 37.98	16.18 ± 3.42	236.13 ± 103.33	5.41 ± 2.24	3.76 ± 4.06	5.47^b^ ± 6.84	0.17 ± 0.04	0.17 ± 0.04	0.17 ± 0.04
	GA	115	119.13 ± 48.57	116.69 ± 46.13	18.16 ± 5.42	115.46 ± 46.45	113.24 ± 46.05	16.87 ± 4.64	229.93 ± 91.28	5.83 ± 2.33	4.27 ± 4.45	6.54^a^ ± 5.75	0.16 ± 0.06	0.17 ± 0.06	0.16 ± 0.05
	GG	187	108.9 ± 45.83	107.98 ± 44.28	17.39 ± 4.64	108.93 ± 44.6	106.61 ± 44.36	16.67 ± 4.4	214.6 ± 88.07	5.73 ± 2.39	3.15 ± 2.89	4.49^b^ ± 3.82	0.18 ± 0.07	0.18 ± 0.07	0.18 ± 0.07
ANOVA P			0.179	0.258	0.413	0.472	0.436	0.896	0.298	0.863	0.132	0.002	0.415	0.292	0.177
Kruskal test P			0.249	0.313	0.853	0.554	0.567	0.942	0.398	0.873	0.503	0.004	0.419	0.463	0.368

^1^ left testicular volume; ^2^ left testicular weight; ^3^ left epididymis weight; ^4^ right testicular volume; ^5^ right testicular weight; ^6^ right epididymis weight; ^7^ total testicular weight; ^8^ testicular index; ^9^ variation coefficient between left and right testis weight; ^10^ variation coefficient between left and right epididymis weight; ^11^ right epididymis weight/ right testis weight; ^12^ left epididymis weight/ left testis weight; ^13^ total epididymis weight/ total testis weight.

**Table 5 animals-10-00740-t005:** Association analysis between different haplotypes and testicular traits.

Haplotype	N	LTV ^1^	LTW ^2^	LEW ^3^	RTV ^4^	RTW ^5^	REW ^6^	TTW ^7^	TI ^8^ (‰)	VCTW ^9^ (%)	VCEW ^10^ (%)	REW/ RTW ^11^	LEW/ LTW ^12^	TEW/ TTW ^13^
CTTTCTGAGAGA	64	119.38 ± 53.49	115.67 ± 49.85	18.39 ± 5.78	116.08 ± 48.25	112.81 ± 48.49	16.74 ± 4.45	228.48 ± 97.57	5.93 ± 2.55	4.46 ± 4.44	7.37^a^ ± 6.83	0.16 ± 0.06	0.17 ± 0.06	0.17 ± 0.05
CTTTTTAAAAGA	27	119.26 ± 49.61	117.87 ± 48.84	16.66 ± 4.94	116.81 ± 51.62	114.57 ± 49.53	15.72 ± 5.46	232.44 ± 97.73	5.53 ± 2.52	3.48 ± 3.75	5.71^ab^ ± 4.28	0.15 ± 0.05	0.16 ± 0.05	0.15 ± 0.05
TTTTCCGGGGGG	79	104.68 ± 39.75	104.03 ± 38.48	17.49 ± 4.55	104.49 ± 38.85	103.09 ± 38.09	16.95 ± 3.95	207.11 ± 75.91	5.65 ± 2.11	3.39 ± 3.03	4.52^b^ ± 3.82	0.18 ± 0.07	0.19 ± 0.07	0.18 ± 0.06
TTTTCTGAGAGG	56	104.39 ± 43.16	103.91 ± 41.77	16.95 ± 4.63	106.36 ± 43.18	102.3 ± 43.97	16.16 ± 4.6	206.21 ± 85.16	5.40 ± 2.42	2.89 ± 3.09	4.66^b^ ± 4.15	0.18 ± 0.07	0.19 ± 0.08	0.18 ± 0.07
Anova P		0.209	0.327	0.360	0.448	0.445	0.435	0.381	0.666	0.195	0.009	0.257	0.22	0.104
Kruskal test P		0.337	0.453	0.373	0.548	0.618	0.309	0.506	0.687	0.319	0.030	0.118	0.205	0.125

^1^ left testicular volume; ^2^ left testicular weight; ^3^ left epididymis weight; ^4^ right testicular volume; ^5^ right testicular weight; ^6^ right epididymis weight; ^7^ total testicular weight; ^8^ testicular index; ^9^ variation coefficient between left and right testis weight; ^10^ variation coefficient between left and right epididymis weight; ^11^ right epididymis weight/ right testis weight; ^12^ left epididymis weight/ left testis weight; ^13^ total epididymis weight/ total testis weight.

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
