# Peer review of "Expression of DAZL Gene in Selected Tissues and Association of Its Polymorphisms with Testicular Size in Hu Sheep"

_animals, 2020, doi:10.3390/ani10040740_

Round 1

Reviewer 1 Report

The manuscript submitted by Luo et al. analyzed the expression profiles and SNPs of a famous candidate gene related to male fertility (DAZL) in sheep, which can be useful for ram selection at early stage. The sample size they analyzed is adequate to screen SNPs in DAZL, and the results gave indicates that this gene maybe important for testicular development. The manuscript is generally well written, it could be accepted for publication after a minor revision: (1) In table 1, some primers should amplify partial of exon, not the whole exon, please clarify that. (2) Delete “gene” in line 119. (3) The authors gave some calculated traits in statistical analysis part, the authors need to explain the reason they to do so, and what usage in the evaluation of ram fertility, in Discussion part. (4) How can author make sure Hu sheep are mature at six month, did thy do any semen testing?

Reviewer 2 Report

An interesting paper exploring expression profiles of DAZL gene and gene  SNP polymorphisms with testicular parameters in Hu sheep of China. My comments are focused mainly on statistical analysis that needs further clarifications or even re-analyses.

Statistical analysis

Figures 1 and 2 present results of means comparisons of relative expression data, with, however, no indication of which statistical test has been employed. It is quite usual such data not to follow normality, so I guess, some kind of non-parametric test must have been applied during this analysis. Pls explain in more detail.

lines 155-156. Pls be noted that all derived parameters presented here are ratio variables and it may reasonably be expected that some (or all) of them deviate from normality. This may also be the case for some of the rest single variables. Pls explain whether you have tested for data normality. If normality is not the case, then statistical analysis is not correct. Furthermore, how have you dealt with the multiple comparison problem? Have you applied any correction method for means comparison? (you have 3 genotypes).  Another important issue here relates to multi-collinearity arising from high LD levels between SNPs. Estimates of genetic effects for multiple, highly correlated variants residing on the same haplotype block are expected biased because the effect(s) are ‘shared’ by many markers. Under such conditions, it is vital to have a parsimonious model involving limited number of regressors (SNPs). To this end, application of the LASSO (Least Absolute Shrinkage and Selection Operator) technique or the use of only one representative SNP (that is highly correlated with the rest) may serve a good solution.

line 160: what is an adjusted linear model?

lines 164-166. Association analysis between haplotypes and testis should also account for the rest fixed effects (e.g. age). However, you mention here that only one-way (I guess haplotype effect) ANOVA was performed. Also be noted that some haplotypes are not adequately represented (n=8 to 11).

Results

Table 4. Pls provide number of genotypes. 

line 207-209. Since SNPs show high LD levels, HWE for one SNP is reasonably be expected for the rest SNPs as well.

lines 216-222 and text throughout. The wording 'significant tendency association' describes an aspect of the result that doesn’t exist since there is no ‘trend’, in any direction. Results are either significant or not and can’t be qualified. 

Reviewer 3 Report

Please find my comments in the attached file.
